# Efficacy of Virtual Reality Simulation in Teaching Basic Life Support and Its Retention at 6 Months

**DOI:** 10.3390/ijerph20054095

**Published:** 2023-02-24

**Authors:** Jordi Castillo, Encarna Rodríguez-Higueras, Ricardo Belmonte, Carmen Rodríguez, Alejandro López, Alberto Gallart

**Affiliations:** Departament Infermeria, Universitat Internacional de Catalunya (UIC), Sant Cugat del Vallès, 08195 Barcelona, Spain

**Keywords:** simulation, virtual reality, cardiopulmonary resuscitation, education

## Abstract

Educational efficiency is the predetermining factor for increasing the survival rate of patients with cardiac arrest. Virtual reality (VR) simulation could help to improve the skills of those undergoing basic life support–automated external defibrillation (BLS–AED) training. Our purpose was to evaluate whether BLS–AED with virtual reality improves the skills and satisfaction of students enrolled in in-person training after completing the course and their retention of those skills 6 months later. This was an experimental study of first-year university students from a school of health sciences. We compared traditional training (control group—CG) with virtual reality simulation (experimental group—EG). The students were evaluated using a simulated case with three validated instruments after the completion of training and at 6 months. A total of 241 students participated in the study. After the training period, there were no statistically significant differences in knowledge evaluation or in practical skills when assessed using a feedback mannequin. Statistically significant results on defibrillation were poorer in the EG evaluated by the instructor. Retention at 6 months decreased significantly in both groups. The results of the teaching methodology using VR were similar to those obtained through traditional methodology: there was an increase in skills after training, and their retention decreased over time. Defibrillation results were better after traditional learning.

## 1. Introduction

Data from 27 European countries collected in 2016 showed that the incidence of cardiac arrest (CA) ranged between 19 and 104 cases per 100,000 inhabitants per year. Although 25% of these patients arrived at the hospital with a pulse, only 10.3% survived 30 days after hospital discharge. CA continues to be a large public health problem [1].

Although the population is becoming increasingly aware of and trained in basic life support (BLS) and the use of an automated external defibrillator (AED), fewer than half of cardiac arrests are attended by eyewitnesses and only a minority of CA cases receive adequate application of these techniques from first responders. Cheng et al. [2] suggested a deepening of educational efficiency and its implementation in the general population, especially in students and basic education centers, relatives of those at risk of CA, etc. This is one of the predetermining factors for increasing patient survival rates. In the same sense, a group of experts of the American Heart Association (AHA) concluded that one of the limitations of the 2015 guidelines [3] was the scarce optimization of educational strategies, and this gap continues in 2021 [4].

From the current educational strategies in teaching BLS–AED to traditional training with an instructor as a model, recently, alternative options (mixed or online) have emerged [5], with controversial results due to the progressive decline of knowledge and skills [6] resulting in suboptimal clinical care and poor survival outcomes [2]. It is important to develop new teaching techniques to update and maintain knowledge and skills in BLS–AED [7]. Virtual reality (VR) is one such option, with numerous publications since 2014 related to the training in BLS–AED [8], most of which are focused on student satisfaction [9], as well as instructors [10,11] or observational studies [12].

The evaluation of new educational techniques must also include the economic costs, the simplification of equipment, student facilities, especially those related to updating and maintaining knowledge, and finally the improvement in clinical practice with a reduction in morbimortality.

For VR to be introduced and recommended as a teaching methodology, its effectiveness must be validated. The main objective of our study was to evaluate whether training in BLS–AED with VR improves the skills and satisfaction of health sciences students with respect to traditional training immediately after the course and their retention at 6 months.

## 2. Materials and Methods

We conducted an experimental study of teaching innovation in BLS–AED that compared the results between a control group, CG (traditional training), and an experimental group, VG (training with VR). The study was approved by the Ethics Committee of the University.

The population consisted of first-year university students from the School of Health Sciences (Medicine, Nursing and Psychology) in the 2020–2021 academic year.

A representative sample presented with “non-inferiority analysis” and was obtained from the study conducted by the same group of researchers to compare 2 teaching methodologies in BLS–AED [13], where a minimum of 40 students per group was sufficient to obtain conclusive results after the training and at 6 months.

Students who had attended a BLS course in the last 3 years or did not sign the informed consent form were excluded. The students were randomized with QuickCalcs software (https://www.graphpad.com/quickcalcs/ accessed on 6 September 2020).

The experimental and traditional training randomized groups received official training from the European Resuscitation Council (ERC) based on a 4-step methodology [14] of 4 teaching hours modified by the training recommendations of the COVID-19 pandemic (avoid forehead–chin maneuver and approach to assess breathing, and minimize the risk of infection by placing a cloth or towel over the mouth of the mannequin before performing chest compressions and defibrillation) [15].

The classrooms were large, well ventilated and adequately disinfected before and after the training. The instructor–student ratio was 1:6. Each student worked with a Little Anne Laerdal^®^ (Laerdal, Madrid, Spain) mannequin with feedback.

The outline of the course is shown in Table 1. For the integrated simulations, 6 cases were used that were completely solved on an individual basis in a 15 min period. The rest of the students observed the performance of their classmates or practiced on their own mannequins.

The two groups received the same training, with the only difference being that the integrated simulations were performed in the virtual group with the help of VR. The SVB-AED training tool in VR was developed by the company LUDUS^®^ under scientific supervision. Designed and implemented for use in VR, the application allows interaction with the 3D virtual environment with six degrees of freedom (6DoF). The interaction with the tangible elements of the virtual environment is performed with “leap motion”, technology that monitors the movement of the hands. The software is compatible with HTC Vive PRO and Cosmos; Oculus Rift; and HP Reverb. The instructor began by explaining the use of the platform, the placement of the glasses, the permitted movements and the environment. The software contains six scenarios (one per student, identical to the CG), and once explained by the instructor, all students could perform BLS maneuvers on their mannequin or watch their classmate using VR complete the maneuver on a big screen. The simulation ended with a structured debriefing [16,17]. Each simulation lasted 15 min.

Prior to the course, students completed a sociodemographic questionnaire (age, sex, faculty, etc.), estimating their knowledge and skills in BLS–AED prior to training, self-assessed with a Likert scale of 0–10 and their impression of the training ability of the course. To assess the degree of satisfaction with the course, a 10-item questionnaire with a Likert scale (0–10) was created, including an open-ended question to determine students’ opinions of the new training methodology.

The students were evaluated through a simulated case using 3 instruments after the training and after 6 months:For the evaluation of theoretical knowledge, a multiple-choice question (MCQ) was used by the local scientific society which provides accreditation to students participating in BLS courses. This MCQ is used in different published articles [13,18].For the evaluation of practical skills, which was conducted by the instructor, a validated grid was used in the process of publication with acceptable psychometric properties (Cronbach’s alpha of 0.72 and a KMO statistic value of 0.719) to score the practical skills using the BLS–AED algorithm. This grid was adapted to the pandemic situation, replacing the assessment of “see, hear and feel breathing” with the following variables: Put the hand on the patient’s thorax, turn the victim’s head and cover their mouth with a handkerchief (the maximum score is 16 points and the minimum is 0 points).For the evaluation of technical skills, the data monitored by the intelligent QCPR mannequin from Laerdal^®^ on quality cardiopulmonary resuscitation (CPR) were used (the software was configured to evaluate only chest compressions for 2 min).

The qualitative variables are presented as absolute frequencies (n) and percentages (%), and the quantitative variables are presented as the mean and standard deviation (SD) or median and quartile interval (QI). For the statistical comparison between groups, Student’s *t*-test (for independent data) or Mann–Whitney U tests were used according to the normality of quantitative variables, whereas the chi-squared test was used for qualitative variables. For the comparison of means between the values obtained at the end of the course and at 6 months, Student’s t-test was used for paired data. The data were converted to a scale from 0 to 10 to facilitate their interpretation. Statistically significant differences were *p*-values equal to or less than 0.05. For the statistical analysis, the software SPSS for Windows version 18 was used.

## 3. Results

A total of 241 students participated in the study. The sociodemographic, academic and perception variables were homogeneously distributed throughout the two groups (Table 2).

After the training, there were no statistically significant differences in the evaluation of knowledge or in the practical skills evaluated by the mannequin (Table 3). When breaking down these data, this was observed (Table 4).

Competency retention at six months decreased significantly in each of the evaluations performed (Table 3 and Table 4) by values of around 1.5 points out of 10 in theoretical knowledge and practical skills and by more than 10 out of 100 in the overall score calculated by the intelligent dummy. At 6 months, global knowledge and skill scores were similar between the two groups. The only statistically significant value between the two groups was found in the correct position of the hands, evaluated by the smart mannequin, which was higher in the CG (Table 3).

The qualitative satisfaction measurement yields a similar score, with an average of 9.56 points for the 10 questions between the two groups, without significant differences between them (*p* = 0.99). One hundred percent of the VG commented that VR was a very good training tool, and the open comments organized and grouped: emotional realism (30%), scenic realism (34%) and improvement in teaching methodology (found in 31% of the answers).

## 4. Discussion

Since the 1990s, an explosion in the application of virtual environments and related technologies has occurred throughout the healthcare field. Applications of these technologies are being implemented in the following areas: surgical procedures (remote surgery or telepresence, planning and simulation of procedures before surgery), medical therapy, preventive medicine and patient education and medical education and training. Such applications have improved the quality of healthcare, and in the future, they will result in substantial cost savings [4].

Virtual worlds enable the inclusion and practice of activities for experiential learning and the simulation and modeling of complex scenarios, providing opportunities for collaboration and co-creation that cannot easily be experienced through other platforms [12].

Responding to an emergency situation such as a cardiac arrest requires widespread training in BLS–AED for the entire population, especially first responders, relatives of at-risk patients, police officers, firefighters, etc. VR makes scenario creation and collaboration easier and, in the short term, will reduce costs.

Although international guides continue to support traditional face-to-face training in BLS–AED [4], they are beginning to recommend alternative training methodologies such as virtual learning [13], self-training [19], peer tutoring [20], gaming [21] and, over the last decade, VR, the use of which increased due to the recent pandemic [5]. Since the first publications of Sameraro et al. [11], different European organizations have used VR in training and have published their results [22,23,24], although most of the published studies are heterogeneous, with observational designs [12,13].

Our study has the strength of working with a homogeneous and representative sample throughout the study (maintaining a minimum of 40 participants per group at six months), with validated data collection instruments and with a follow-up over time.

In addition, it integrates VR within the traditional classroom methodology (for the VG), using the teaching methodology endorsed by the ERC [14] in order to demonstrate the effectiveness of VR. The articles found are heterogeneous in terms of the teaching methodology used, comparing 20 min of VR training with the same amount of face-to-face time [24,25] and another 35 min of VR [26], making it difficult to compare teaching methodologies. Others interpret VR exclusively as online training [5] without comparison between groups. Regardless of the optimal duration, in our opinion, the most important thing is that the system manages to motivate the student, capture their attention and improve their learning [27].

The objective evaluation of skills acquired during training is also heterogeneous in the literature: some use MCQ [11,28], whereas others use mannequin data [23,24], and still others use Likert-type scales to evaluate satisfaction [12]. However, none of them use all the current tools (objective and validated) to ensure the competence of the students in BLS–AED.

After training, knowledge from VR was similar to that obtained from traditional training [8.21 (1.41) vs. 8.44 (1.65); *p* = 0.24], with an improvement after training in both groups similar to other “blended” methodologies [13] used in the literature. We can confirm that VR has no influence on the acquisition of knowledge in BLS–AED.

In the skills evaluated by the mannequin, no significant differences were observed between the two groups [67.86 (24.99) vs. 64.54 (28.85); *p* = 0.34]. However, we obtained depth compression results 0.5 mm lower than those obtained in the study by Beom et al. [29] in both groups. Their study was aimed at professional ambulance technicians, which could explain this difference. On the other hand, skills evaluated by the instructor during the simulated case were statistically better after training in the CG [9.10 (1.2) vs. 8.61 (1.48); *p* = 0.05], mainly influenced by the item “Safe Discharge”. We believe that these isolated data have little value both in this study and in the clinic.

The satisfaction obtained in this study is similar to that found in the literature but grouped into three large blocks: the scenic realism that it offers [9] with statements such as, “I think that VR brings that sensation of feeling that it is happening to you” and “It brings an experience, since it is as if you lived the case without imagining it”; realism or emotional security [28], with statements such as, “It provides the assurance that you will know how to react in a similar situation”; or how to improve the teaching methodology [30] with statements such as, “More effective learning” and “You feel much more effective, I think that it is the part in which you learn the most”. We believe that VR is capable of taking a qualitative leap in the training of BLS–AED. We believe that simulation is a good tool which allows students to acquire and practice skills without risk of injury or harm to the patient [31]. However, we question whether it is capable of generating the stress and emotional burden that VR produces in these cases [28]. Although we did not conduct a survey on instructors’ satisfaction and opinion with regard to VR, all of them believe that any initiative to improve the satisfaction and motivation of the students should be introduced, to the greatest extent possible, within the training in BLS–AED.

Like Smith and Hamilton [32], who support the use of VR simulation as a supplemental tool for teaching students, we consider that VR simulation is a BLS–AED teaching method with similar results compared to traditional face-to-face training. Regardless, Cerezo et al. [27] found that the results are discreetly better. We believe, however, that it provides emotional, situational and reality qualities that no other methodology provides and should be introduced for future generations, in training aimed at teenagers and the general population.

It should be noted that VR training in BLS–AED is still in need of improvement; therefore, it is necessary to dedicate time to the students to explain the operation of the actions that must be carried out for the student to become used to the format. The adaptation to glasses was not well tolerated by all students (in our study, four students became dizzy when using VR), so we think that it is necessary that an instructor trained in VR be dedicated exclusively to students who perform this training and that another instructor is needed to guide the rest of the students. The dynamism of the class could be increased by providing more than one set of VR glasses. Both solutions increase the costs of the courses.

Much more development of technology is needed, not only in terms of cabling, hardware, etc., but also to allow more space for student movement, ensuring that the environment is dynamic (in our case, the AED always appeared behind the virtual element that interacted to save the victim, generating distortion. We believe that this is the reason why there is a difference from the control group) and that the people who appear in action interact with each other and the resuscitator to achieve greater realism and participation [23].

VR must be able to immediately evaluate the student without an instructor’s input. The software itself must have a training mode and an evaluation mode.

In addition to improvements in educational performance, it is necessary to demonstrate that these new teaching initiatives are effective in the clinical field. This could be a future research direction.

Six months after the course, the three overall evaluations decreased substantially with respect to the evaluation performed after the training. We have not found articles that measure retention with the use of VR, although this trend is similar to the published literature comparing other methodologies [13]. The introduction of VR did not improve the retention of skills in BLS–AED over time, and hand position was significantly different in the CG [96.38 (16.2) vs. 91.7 (25.3): *p* = 0.02] when we further examined the evaluation performed by the mannequin. These data were excellent in both groups (greater than 90%).

We must continue to search for a method that obtains more sustained retention over time in this discipline, with VR being a good tool for refresher courses. New technologies, which are accessible to the majority of the population, have great potential to provide society with fast, easy and accessible CPR training that can be performed at home at a low cost [24]. As Semeraro et al. [23] argue, we believe that VR in BLS–AED is a valid and acceptable tool for training programs aimed at general populations, schoolchildren and health professionals, with a gaming approach that is very useful for refreshing knowledge.

## 5. Conclusions

The results of the teaching methodology with VR are similar to those of the traditional methodology: there is an increase in competency after training, and its retention decreases over time.

VR has similar results to the integrated simulations within a standard BLS–AED course.

VR technology has room for improvement, and we hope that it will be part of the training in BLS–AED as another tool in the immediate future.

## Figures and Tables

**Table 1 ijerph-20-04095-t001:** Training in traditional BLS–AED or with virtual reality.

		ERC Standard Course	Virtual Reality Course
TIME		Reading the manual	
2 h	15 min	Presentation 15 min	
	15 min	Demonstration 15 min	
	90 min	BLS–AED practice. Skills 4 stages.	
2 h	90 min	Integrated simulations	Virtual reality simulations
	15 min	RP and FBAO	
	15 min	Practical and theoretical evaluation	

ERC; European Resuscitation Council; BLS–AED: basic life support–automated external defibrillation; RP: recovery position; FBAO: foreign body airway obstruction.

**Table 2 ijerph-20-04095-t002:** Sociodemographic characteristics of the participants.

Age (years)		CG (n = 116)	EG (n = 125)	p
		19.84 (4.99)	19.21 (2.49)	0.23
Sex	Female	91 (78.4%)	89 (71.2%)	0.22
	Male	25 (21.5%)	36 (28.8%)	
Faculty	Nursing	47 (40.5%)	48 (38.4%)	0.73
	Medicine	46 (39.7%)	47 (37.6%)	
	Psychology	23 (19.8%)	30 (24%)	
Weight (kg)		60.8 (10.78)	59.1 (12.01)	0.26
Height (cm)		162.28 (29.26)	167.93 (14.56)	0.87
Knowledge perception (0–10)		6.44 (1.64)	6.33 (1.66)	0.61
Skills perception (0–10)		5.34 (1.95)	5.1 (2)	0.36
Course learning		1 (0)	0.98 (0.15)	0.09

CG: control group (traditional training); EG: experimental group (virtual reality simulation).

**Table 3 ijerph-20-04095-t003:** Evaluation of knowledge and skills before and after the course and retention at 6 months.

	At Course Completion			At 6 Months		
	CG (n = 116)	EG (n = 125)	*p*	CG (n = 56)	EG (n = 64)	*p*
Knowledge						
MCQ (0–10)	8.21 (1.41)	8.44 (1.65)	0.24	6.55 (1.56)	6.42 (1.54)	0.75
Skills instructor						
Overall Score (0–10)	9.10 (1.2)	8.61 (1.48)	0.05	6.23 (2.09)	6.25 (2.14)	0.46
Skills Mannequin						
Overall Score (%)	67.86 (24.99)	64.54 (28.85)	0.34	53.7 (32.19)	49.66 (36.7)	0.13
Correct hand positioning (%)	97.73 (11.03)	97.68 (9.94)	0.97	96.38 (16.2)	91.7 (25.3)	0.02
Median depth (mm)	47.1 (7.27)	45.98 (7.70)	0.24	44.71 (8.86)	42.66 (9.73)	0.33
Complete re-expansion (%)	70.52 (34.06)	71.56 (32.28)	0.8	79.53 (33.49)	77.26 (29.61)	0.57
Correct compressions (%)	43.4 (35.99)	41.14 (34.66)	0.62	35.04 (37.79)	32 (34.16)	0.86
Frequency 100–120 (%)	61.86 (30.6)	60.33 (34.94)	0.71	52.18 (35.84)	50.12 (36.34)	0.86

CG: control group (traditional training); EG: experimental group (reality virtual simulation); MCQ: multiple choice question *p*-values < 0.05 are considered statistically significant. Significant values are highlighted in bold.

**Table 4 ijerph-20-04095-t004:** Values obtained from the practical evaluation by the instructors.

		After Training			6 Months after Training		
		CG (n = 116)	EG (n = 125)	*p*	CG (n = 56)	EG (n = 64)	*p*
Assessment of consciousness	Neither shouts nor shakes it	3 (2.6%)	9 (7.2%)	0.15	6 (10.7%)	6 (9.4%)	0.97
	Yells or shakes it	112 (96.6%)	116 (92.8%)		18 (32.1%)	21 (32.8%)	
	Yells and shakes it	1 (0.9%)	0 (0%)		32 (57.2%)	37 (57.8%)	
Assessment of respiration	No hands on thorax or handkerchief in mouth	0 (0%)	4 (3.2%)	0.13	28 (50%)	27 (42.2%)	0.65
	Hands on thorax or handkerchief in mouth	29 (25%)	34 (27.2%)		17(30.4%)	21 (32.8%)	
	Hands on thorax and handkerchief in mouth	87(75%)	87 (69.6%)		11 (19.6%)	16 (25%)	
Respiration assessment time	<2 s	2 (1.7%)	10 (8%)	0.08	18 (32.1%)	31 (48.4%)	0.11
	>10° < 5 s	25 (21.6%)	24 (19.2%)		23 (41.1%)	16 (25%)	
	Between 5 and 10 s	89 (76.7%)	91 (72.8%)		15 (26.8%)	17 (26.6%)	
Asks for the AED	Yes (at some point)	113 (97.4%)	116 (92.8%)	0.24	24 (42.8%)	15 (23.5%)	0.06
	No (at no time)	3 (2.6%)	9 (7.2%)		32 (57.2%)	49 (76.5%)	
Calls 112	Does not call or does not place hands-free device	1 (0.9%)	3 (2.4%)	0.53	7 (12.5%)	5 (7.8%)	0.14
	Calls or places hands-free device	7 (6%)	10 (8%)		2 (7.8%)	0 (0%)	
	Calls and places hands-free device	108 (93.1%)	112 (89.6%)		47 (83.9%)	59 (92.2%)	
AED Pads	Does not place patches according to guidelines	2 (1.7%)	9 (7.2%)	0.1	13 (23.2%)	24 (37.5%)	0.06
	Does apply patches according to guidelines	114 (98.3%)	116 (92.8%)		43 (76.8%)	40 (62.5%)	
Safe discharge	Does not look or verbalize	9 (7.8%)	19 (15.2%)	0.04	14 (25%)	19 (29.7%)	0.84
	Verbalizes or looks	24 (20.7%)	35 (28%)		21 (37.5%)	23 (35.9%)	
	Verbalizes and looks	83 (71.6%)	71 (56.8%)		21 (37.5%)	22 (34.4%)	
Immediate compressions	Does not compress immediately	1 (0.9%)	1 (0.8%)	0.23	9 (16.1%)	14 (21.9%)	0.3
	Compresses when indicated by the AED	30 (25.9%)	45 (36%)		35 (62.5%)	31 (48.4%)	
	Compresses before the AED indicates	85 (73.3%)	79 (63.2%)		12 (21.4%)	19 (29.7%)	
Overall Instructor Note		9.10 (1.2)	8.61 (1.48)	0.05	6.23 (2.09)	6.25 (2.14)	0.46

CG: control group (traditional training); EG: experimental group (reality virtual simulation); AED: automated external defibrillation. *p*-values < 0.05 are considered statistically significant. Significant values are highlighted in bold.

## Data Availability

No new date base was created.

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
