# Peer review of "Efficacy of Virtual Reality Simulation in Teaching Basic Life Support and Its Retention at 6 Months"

_ijerph, 2023, doi:10.3390/ijerph20054095_

Round 1

Reviewer 1 Report

Thank you for allowing me to review this manuscript. The potential of the research is potentially far-reaching for Learning and Teaching communities. I have made extensive comments on the attachment. Once the noted concerns, essentially clarity of writing addressed, and the limitations of the study have been included, the paper will be far more engaging for the reader.

I wish you every success with your progress.

Author Response

Please thank the reviewers for their wise comments. We have learned very much from them.

We have realized near all the corrections suggested by the four reviewers. The reviewers have detected many hyphenated words that surprisingly were not in the initial text. They have all been corrected

Thank you very much

Yours sincerely

Response to Reviewer 1 Comments

Reviewer 1: Line 54: This sentence requires a supporting reference.

Author answer: We cannot support it with a reference because this is the justification of our study.

Reviewer 1: Line 70: What do you mean by this statement? Students who had completed the course in the last 3 years.

Author answer: it means that the students who had attended a BLS course in the last 3 years were excluded of the study.  We have changed the sentence by: “students who had have attended a BLS course in the last 3 years…..”

Reviewer 1: Line 73. Which two groups? Professional groups or randomized groups be clear in your writing

Author answer: we refer to randomized groups. We have added the word “randomized”

Reviewer 1: Line 89-90: under the scientific supervision of IMQ. IMQ what is this acronym mean?

Author answer: We don’t exactly know. It is a scientific instructor of the enterprise who supervises the device functioning. We have delated the acronym because we understand that it is confusing.

Reviewer 1: Line 94: If this information is valuable to the research (not a 'red herring') it needs writing in full with an explanation

Author answer: It is not a “red herring”. These are the technical specifications of the device. We would leave it like this because some readers might be interested in it.

Reviewer 1: Line 97. could perform what?

Author answer: BLS manoeuvers. We have added in the text

Reviewer 1: Line 101. Knowledge and skills in what???

Author answer: In BLS-AED. We have added in the text

Reviewer 1: Line 109. that accredits the courses

Author answer: “which provides accreditation to students participating in BLS courses”. We have included in the text with the aim to make clear which society we are speaking about.

Reviewer 1: suggestion: numbers instead of dots.

Author answer: we have changed dots for numbers.

Reviewer 1:  Line 137. The instructor was evaluated as worse????

Author answer: “that the evaluation of the instructor regarding administration of safe defibrillation was significantly worse (p = 0.05) in the VG”.

We mean that the safe defibrillation was worse in the Virtual Group (VG). The instructor was not worse.

Reviewer 1:  Line 139. Retention of what???

Author answer: we have added at the beginning of the sentence: Competencies (knowledge and skills)

Reviewer 1:  Knowledge and skill scores of what?

Author answer: Competencies in BLS. We have not changed nor added nothing because we said it in line 139.

Reviewer 1:  Evidence to support statement required

Line 167: “Such applications have improved the quality of healthcare, and in the future, they will result in substantial cost savings”.

Author answer:  International guidelines try to gather all the evidence published on BLS studies with the aim to improve the quality of healthcare. We support with the international guidelines reference.

Reviewer 1:  Evidence to support statement required. Line 169-170

Author answer:  Reference number 14 supports adequately the statement

Reviewer 1:  should this be ' the acquisition of knowledge in BLS-AED?' otherwise you are making a very general assumption without testing...

Line 203….. We can confirm that VR has no influence on the acquisition of knowledge

Author answer:  Thank you for your remark. Of course we meant in BLS-AED.

Reviewer 1:  what skills???

We believe that VR is capable of taking a qualitative leap in the training of BLS-AED. We believe that simulation is a good tool which allows students to acquire and practice skills without risk of injury or harm to the patient [29].

Author answer:  All those skills without risk or injury or harm to the patient.

Reviewer 2 Report

You need to review some spellings errors such as words joined by hyphens.

In the discussion rewrite the paragraph from line 234 to 242

The content seems to me to be current and of quality. It tries to demonstrate if the new virtual reality methodologies provide improvements with respect to the acquisition of knowledge and its permanence over time.
Although there are more and more contributions in VR and CPR. The authors complete a gap  and try to answer the question if this methodology improves traditional teachings, in learning and in permanence of knowledge. This opens new avenues of research in different learning modalities, from undergraduate to profesional.
The wording of lines 234 to 240 seems very telegraphic and breaks the common thread of the paragraph. Being the correct contents, the style should be improved.
The bibliography is correct, coherent with the subject and current. Althouht I miss some bibliography about debriefing (line 99) .
The conclusions are consistent with the objectives set. Some details to add

  1. I have signaled the writing mistaque in the paper, I am going to highlight in yelow colour  the words joined by hyphens so that you can rewrite them correctly
  2. Line 35 the reference 2 is duplicated
  3. Line 73. Instead of both groups, experimental and traditional training
  4. I need an explanation about l Line 94 : HTC Vive PRO and Cosmos; Oculus Rift; and HP
  5. Tables are structured and clear .
  6. Line 154 Close parentheses

Author Response

Please thank the reviewers for their wise comments. We have learned very much from them.

We have realized near all the corrections suggested by the four reviewers. The reviewers have detected many hyphenated words that surprisingly were not in the initial text. They have all been corrected

Thank you very much

Yours sincerely

Response to Reviewer 2 Comments

Reviewer 2: The wording of lines 234 to 240 seems very telegraphic and breaks the common thread of the paragraph. Being the correct contents, the style should be improved.

It should be noted that VR training in BLS-AED is still in need of improvement.

It is necessary to dedicate time to the students to explain the operation of the ac-tions that must be carried out for the student to get used to the format.

Adaptation to glasses was not well tolerated by all students (in our study, four students became dizzy when using VR).

It is necessary that an instructor trained in VR be dedicated exclusively to stu-dents who perform this training and that another instructor is needed to guide the rest of the students. The dynamism of the class could be increased by providing more than one set of VR glasses. Both solutions increase the costs of the courses.

Author Answer:  We agree with the reviewer so we have improved the style with some connectors.

It should be noted that VR training in BLS-AED is still in need of improvement, therefore it is necessary to dedicate time to the students to explain the operation of the actions that must be carried out for the student to get used to the format. The adaptation to glasses was not well tolerated by all students (in our study, four students became dizzy when using VR), so we think that it is necessary that an instructor trained in VR be dedicated exclusively to students who perform this training and that another instructor is needed to guide the rest of the students. The dynamism of the class could be increased by providing more than one set of VR glasses. Both solutions increase the costs of the courses.

Reviewer 2: The bibliography is correct, coherent with the subject and current. Althouht I miss some bibliography about debriefing (line 99).

Author Answer: thank you for your suggestion, although when adding a couple of references, I have modified the rest of references in the text

Reviewer 2:  Line 35 the reference 2 is duplicated

Author Answer:  We have deleted the first one.

Reviewer 2:  Line 73. Instead of both groups, experimental and traditional training

Author Answer:  Changed.

Reviewer 2:  I need an explanation about l Line 94 : HTC Vive PRO and Cosmos; Oculus Rift; and HP

Author Answer:  We don’t understand either. Another reviewer has asked the same. These are the technical specifications of the device. We would leave it like this because some readers might be interested in it.

Reviewer 2:  Line 154 Close parentheses

Author Answer:  Changed

Reviewer 3 Report

Thank you for the opportunity to be able to review this interesting ms. on the educational value of VR training in BLS classes. The design of the study is well done, nevertheless, I have some suggestions which may help to further improve the ms.

1) It has been shown that following quality indicators are indicative of a professional thorax compression: a) no interruption b) compression rate of 100-120/min c) recoil d) deepness of compression. I am not convinced whether these quality indicators have been used to assess the quality of the educational groups immediately after and/or 6months afterwards. Could this suggestion be implemented.

2) There are multiple comparisons performed without adequate corrections. I would suggest that a clear sample size calculation is presented ("non-inferiority analysis") with a defined main endpoint. This will help to increase the quality of the presentation of data.

3) The discussion section is a bit lengthy and could be shortened. In addition, the authors should check English language and the writing of the discussion section. Currently, there are a lot of sentences standing alone (e.g. at the end of the discussion paragraph). It should be written to allow fluent reading.

Author Response

Please thank the reviewers for their wise comments. We have learned very much from them.

We have realized near all the corrections suggested by the four reviewers. The reviewers have detected many hyphenated words that surprisingly were not in the initial text. They have all been corrected

Thank you very much

Yours sincerely

Response to Reviewer 3 Comments

Reviewer 3: It has been shown that following quality indicators are indicative of a professional thorax compression: a) no interruption b) compression rate of 100-120/min c) recoil d) deepness of compression. I am not convinced whether these quality indicators have been used to assess the quality of the educational groups immediately after and/or 6months afterwards. Could this suggestion be implemented.

Author answer: These are internationally recognized quality indicators for good cardiac compression.

Reviewer 3: There are multiple comparisons performed without adequate corrections. I would suggest that a clear sample size calculation is presented ("non-inferiority analysis") with a defined main endpoint. This will help to increase the quality of the presentation of data.

Author answer: I agree with you and we have included this concept in the text

Reviewer 3: The discussion section is a bit lengthy and could be shortened. In addition, the authors should check English language and the writing of the discussion section. Currently, there are a lot of sentences standing alone (e.g. at the end of the discussion paragraph). It should be written to allow fluent reading.

Author answer: Reviewer 2 suggested the same and we have written in a way to allow fluent reading. In regards to English language, the whole text has undergone English language editing by MDPI.

Reviewer 4 Report

This manuscript described that virtual reality (VR) simulation was equally effective for teaching basic life support as classical instructor assisted teaching method. This manuscript is well constructed and findings give us new evidences. A little concern is following.

1, There are many unnecessary hyphens in this article. Please check them up and use correct style.

2, When we teaching AED session using mannequin, to teach jerking movement following shock delivery has been very difficult. These jerking movement could be showed using VR simulation. I wish some real VR images during AED session could be presented as figure if possible.

3, Were there any chances for medical students or nurses to see or to attend real resuscitation scene before checking retention? That may influence results of retention.

Author Response

Please thank the reviewers for their wise comments. We have learned very much from them.

We have realized near all the corrections suggested by the four reviewers. The reviewers have detected many hyphenated words that surprisingly were not in the initial text. They have all been corrected

Thank you very much

Yours sincerely

Response to Reviewer 4 Comments

Reviewer 4: When we teaching AED session using mannequin, to teach jerking movement following shock delivery has been very difficult. These jerking movement could be showed using VR simulation. I wish some real VR images during AED session could be presented as figure if possible.

Author answer: We have a lot of real VR images during the training course but unfortunately we don’t have any image during AED session.

Reviewer 4: Were there any chances for medical students or nurses to see or to attend real resuscitation scene before checking retention? That may influence results of retention.

Author answer: it is highly unlikely that students will be able to attend an actual cardiac arrest before the six-month retention assessment as they do not begin internships in hospitals until the second year
